

# Measurement of interaction-dressed Berry curvature and quantum metric in solids by optical absorption

Wei Chen⋆ and Gero von Gersdorff.

Department of Physics, PUC-Rio, Rio de Janeiro 22451-900, Brazil

⋆ wchen@puc-rio.br

## Abstract

The quantum geometric properties of a Bloch state in momentum space are usually described by the Berry curvature and quantum metric. In realistic gapped materials where interactions and disorder render the Bloch state not a viable starting point, we generalize these concepts by introducing dressed Berry curvature and quantum metric at finite temperature, in which the effect of many-body interactions can be included perturbatively. These quantities are extracted from the charge polarization susceptibility caused by linearly or circularly polarized electric fields, whose spectral functions can be measured from momentum-resolved exciton or infrared absorption rate. As a concrete example, we investigate Chern insulators in the presence of impurity scattering, whose results suggest that the quantum geometric properties are protected by the energy gap against many-body interactions.

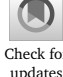
# 1 Introduction

The geometric properties of a quantum state $|\psi(\mathbf{k})\rangle$ in the $D$-dimensional parameter space $\mathbf{k} = (k^1, k^2...k^D)$ has long been of tremendous interest in many areas of physics. The first and perhaps most important aspect of this kind is the Berry phase [1], which is a geometric phase associated with the evolution of the quantum state in a closed trajectory in the parameter space, and is the mechanism behind numerous phenomena such as quantized Hall conductance [2,3] and anomalous velocity [4], just to list a few. The integrand in the calculation of Berry phase is the Berry curvature $\Omega_{\mu\nu}$, which has been measured experimentally in cold atoms [5,6] and solids [7], and is further recognized as the imaginary part of the quantum geometric tensor [8] $T_{\mu\nu}$. The real part of $T_{\mu\nu}$ is yet another important geometric quantity called quantum metric [9] $g_{\mu\nu}$, which has also been measured by means of Rabi oscillations [10]. Generically, how the quantum state $|\psi(\mathbf{k})\rangle$ rotates in the Hilbert space as the parameter changes from $\mathbf{k}$ to $\mathbf{k} + \delta\mathbf{k}$ defines the quantum metric according to $|\langle\psi(\mathbf{k})|\psi(\mathbf{k} + \delta\mathbf{k})\rangle| = 1 - \frac{1}{2} g_{\mu\nu}^{\psi} \delta k^{\mu} \delta k^{\nu}$ [10–24]. This aspect is particularly important to describe quantum phase transitions, since the quantum metric generally diverges near the critical point $\mathbf{k}_c$ regardless of any detail of the system, giving rise to the notion of fidelity susceptibility [25–31].

Despite the ubiquity of Berry curvature and quantum metric behind numerous quantum phenomena, their very definition becomes rather ambiguous in realistic materials subject to many-body interactions and at finite temperature. This situation is relevant to the gapped fermionic systems such as semiconductors, superconductors, or topological insulators that are subject to various complications like disorder, electron-electron, or electron-phonon interactions. In this case, $|\psi(\mathbf{k})\rangle$ is the filled band state at momentum $\mathbf{k}$ [32,33], which however is no longer an energy eigenstate in the presence of many-body interactions, and moreover the state is only partially filled at finite temperature due to Fermi statistics. As a result, one must resort to a more generalized definition for $\Omega_{\mu\nu}$ and $g_{\mu\nu}$. In addition, if the interaction is weak, they must be able to be defined perturbatively, and recover the usual definition in the noninteracting and zero temperature limit.

In this paper, we provide such a generalized formalism for $g_{\mu\nu}$, $\Omega_{\mu\nu}$, and $T_{\mu\nu}$ that are applicable to realistic gapped materials at finite temperature and in the presence of many-body interactions. Our construction is based on the observation that the momentum-derivative in the calculation of $g_{\mu\nu}$ and $\Omega_{\mu\nu}$ actually corresponds to the dipole energy caused by an oscillating electric field [33]. The oscillating field causes the exciton or infrared absorption of the gapped material, and the frequency-integrated absorption rate in the zero temperature and noninteracting limit nicely recovers the usual definition of Berry curvature and quantum metric [17,34,35]. Since the absorption rate itself is a well-defined, experimentally measurable quantity even in the presence of many-body interactions and at finite temperature, it serves as a generalized definition for the Berry curvature and quantum metric. The absorption rate can be formulated within a linear response theory of charge polarization susceptibility, in a way analogous to the theory of exciton absorption rate in semiconductors caused by the minimal coupling between electrons and the vector field [36,37]. Moreover, our formalism introduces the spectral functions for the experimental measurements of Berry curvature and quantum metric, and we will discuss how many-body interactions influence the shape of these spectral functions.

The structure of the paper is organized in the following manner. In Sec. II A, we introduce the linear response theory of charge polarization susceptibility in gapped materials, from which the interaction-dressed Berry curvature and quantum metric naturally emerge. The recovery to the usual definition of Berry curvature and quantum metric in the zero temperature and noninteracting limit is demonstrated explicitly in Sec. II B, and the perturbative calculation of these quantities in the presence of interactions is discussed in Sec. II C. In Sec. II D, we link

the susceptibility to the exciton and infrared absorption rate, thereby providing a concrete measurement protocol for the dressed Berry curvature and quantum metric. In Sec. II E, we use the Chern insulator with impurities as a concrete example to elaborate how the spectral functions are influenced by interactions. Finally, the results are summarized in Sec. III.

## 2 Linear response theory of Berry curvature and quantum metric

### 2.1 Susceptibility formalism for Berry curvature and quantum metric

We begin by recalling that in the noninteracting and zero temperature limit, the quantum geometric tensor, quantum metric, and Berry curvature of a quantum state $|\psi(\mathbf{k})\rangle$ at momentum $\mathbf{k}$ are defined by

$$
\begin{aligned}
T_{\mu\nu}(\mathbf{k}) &= \langle\partial_\mu\psi|\partial_\nu\psi\rangle - \langle\partial_\mu\psi|\psi\rangle\langle\psi|\partial_\nu\psi\rangle, \\
g_{\mu\nu}(\vec{k}) &= \frac{1}{2}\langle\partial_\mu\psi|\partial_\nu\psi\rangle + \frac{1}{2}\langle\partial_\nu\psi|\partial_\mu\psi\rangle - \langle\partial_\mu\psi|\psi\rangle\langle\psi|\partial_\nu\psi\rangle, \\
\Omega_{\mu\nu}(\vec{k}) &= i\langle\partial_\mu\psi|\partial_\nu\psi\rangle - i\langle\partial_\nu\psi|\partial_\mu\psi\rangle,
\end{aligned}
\tag{1}
$$

where $\partial_\mu \equiv \partial/\partial k^\mu$. Our interest is to investigate these quantities for the electrons in gap materials with multiple valence and conduction bands. In the absence of interactions, the electrons in the material are described by a second-quantized fermionic Hamiltonian $H_0(\mathbf{k}) = \sum_{\ell\ell'} h_{\ell\ell'}(\mathbf{k})c^\dagger_{\ell\mathbf{k}}c_{\ell'\mathbf{k}}$, where $\ell$ denotes the degrees of freedom of the fermionic basis like orbitals, spins, etc. After diagonalizing the single-particle Hamiltonian $h(\mathbf{k})|n\mathbf{k}\rangle = E_{n\mathbf{k}}|n\mathbf{k}\rangle$, we introduce the creation operator $c^\dagger_{n\mathbf{k}}$ for the eigenstate $|n\mathbf{k}\rangle$. Some of these $E_{n\mathbf{k}}$'s may be degenerate, such as the spin degeneracy, but this does not affect our formalism. At zero temperature, all the valence band states $E_{n\mathbf{k}} < 0$ are filled and all the conduction band states $E_{n\mathbf{k}} > 0$ are empty. Suppose there are $N_-$ valence bands, then the fully antisymmetric valence band state is [32,33]

$$
|\psi^{\mathrm{val}}(\mathbf{k})\rangle = \frac{1}{\sqrt{N_-!}}\epsilon^{n_1 n_2 \dots n_{N_-}}|n_1\mathbf{k}\rangle|n_2\mathbf{k}\rangle\dots|n_{N_-}\mathbf{k}\rangle,
\tag{2}
$$

which may be inserted into Eq. (2) to obtain the corresponding noninteracting $g_{\mu\nu}$ and $\Omega_{\mu\nu}$. Note that this state is not a physically sensible state in our full Fock space since it ignores all the other momenta $\neq \mathbf{k}$, but the resulting metric and curvature are meaningful and measurable, as elaborated below [32,33].

Our aim is to present a linear response theory that links Berry curvature and quantum metric to optical absorption experiments. In fact, this strategy of formulating Berry curvature in terms of a certain kind of response caused by some external field has been explored in several previous works. Shin et al. consider the charge and spin current caused by an oscillating vector potential, and show that Berry curvature as the anomalous velocity can be extracted from the time-evolution of Bloch states [38]. Gritsev and Polkovnikov elaborate that Berry curvature can be extracted from the response of the generalized force caused by adiabatically quenching a driving parameter, a phenomenon called dynamical quantum Hall effect [39]. Moreover, the quantized Hall conductance, which may be derived from expanding the Bloch state to leading order in the external field [40], can also be expressed in terms of a frequency-derivative of a linear response function at the zero frequency limit [41]. In contrast, our construction links the Berry curvature to the exciton or infrared absorption experiments performed at finite temperature, introduces the Berry curvature spectral function that can incorporate any many-body effects in real materials and be expressed by Feynman diagrams, and moreover elaborates that quantum metric also emerges out of the same linear response theory, as we shall see below.

We now consider the application of an external electric field $E^\mu$ that couples to the operator $i\partial_\mu$, which also plays the role of the generator of the transformation from $\mathbf{k}$ to $\mathbf{k} + \delta\mathbf{k}$ on the momentum space manifold, described by the dipole energy [17, 33, 42]

$$\delta h(\mathbf{k}) = -iqE^\mu \partial_\mu, \tag{3}$$

where $q$ is the charge of the particle. The change of Hamiltonian in the second-quantization formalism is

$$\delta H(\mathbf{k}) = \sum_{nn'} \langle n\mathbf{k}|\delta h(\mathbf{k})|n'\mathbf{k}\rangle \, c_{n\mathbf{k}}^\dagger c_{n'\mathbf{k}} = -qE^\mu U_\mu(\mathbf{k}), \tag{4}$$

which defines the charge polarization operator $U_\mu$

$$U_\mu(\mathbf{k}) = \sum_{nn'} \mathcal{A}_\mu^{nn'}(\mathbf{k}) \, c_{n\mathbf{k}}^\dagger c_{n'\mathbf{k}} = -U_\mu^\dagger(\mathbf{k}),$$

$$\mathcal{A}_\mu^{nn'}(\mathbf{k}) = \langle n\mathbf{k}|i\partial_\mu|n'\mathbf{k}\rangle \equiv \mathcal{A}_\mu^{nn'}, \tag{5}$$

where $\mathcal{A}_\mu^{nn'}$ is the non-Abelian gauge field defined from the eigenstates. In the presence of interactions described by a second-quantized Hamiltonian $H'$, and denoting the unperturbed Hamiltonian of the whole system by $H_0 = \sum_{\mathbf{k}} H_0(\mathbf{k})$, the operators evolve with time according to $U_\mu(\mathbf{k}, t) = e^{i(H_0+H')t} U_\mu(\mathbf{k}) e^{-i(H_0+H')t}$, except $\mathcal{A}_\mu^{nn'}$ which has no dynamics.

The central quantity in our formalism is the susceptibility $\chi_{\mu\nu}$ of the ensemble average of the charge polarization operator $U_\mu$

$$\langle U_\mu(\mathbf{k}, t) \rangle = \chi_{\mu\nu}(\mathbf{k}, t) q E^\nu(t), \tag{6}$$

caused by the application of the electric field $E^\nu(t) = E^\nu e^{-i\omega t}$. Within linear response theory, the Matsubara version of the susceptibility is calculated by

$$\chi_{\mu\nu}(\mathbf{k}, i\omega) = \int_0^\beta d\tau \, e^{i\omega\tau} \chi_{\mu\nu}(\mathbf{k}, \tau) = -\int_0^\beta d\tau \, e^{i\omega\tau} \langle T_\tau U_\mu(\mathbf{k}, \tau) U_\nu^\dagger(\mathbf{k}, 0) \rangle, \tag{7}$$

where $i\omega = \text{integer} \times 2\pi i/\beta$ is the bosonic Matsubara frequency, and the retarded version can be obtained upon an analytical continuation $i\omega \to \omega + i\eta$. We propose the imaginary part of the symmetrized retarded susceptibility to be the quantum metric spectral function, and the real part of the antisymmetrized one to be the Berry curvature spectral function

$$g_{\mu\nu}^d(\mathbf{k}, \omega) \equiv -\frac{1}{2\pi} \text{Im}\left[\chi_{\mu\nu}(\mathbf{k}, \omega) + \chi_{\nu\mu}(\mathbf{k}, \omega)\right],$$

$$\Omega_{\mu\nu}^d(\mathbf{k}, \omega) \equiv -\frac{1}{\pi} \text{Re}\left[\chi_{\mu\nu}(\mathbf{k}, \omega) - \chi_{\nu\mu}(\mathbf{k}, \omega)\right], \tag{8}$$

$$T_{\mu\nu}^d(\mathbf{k}, \omega) \equiv \frac{1}{2\pi}\left[i\chi_{\mu\nu}(\mathbf{k}, \omega) - i\chi_{\nu\mu}^*(\mathbf{k}, \omega)\right],$$

where the superscript $d$ indicates that these quantities are dressed by interactions. The dressed quantum metric, Berry curvature, and quantum geometric tensor to be the integration of the spectral functions over positive frequency, since we aim at capturing the absorption rate

$$\mathcal{O}_{\mu\nu}^d(\mathbf{k}) = \int_0^\infty d\omega \, \mathcal{O}_{\mu\nu}^d(\mathbf{k}, \omega), \tag{9}$$

where $\mathcal{O}_{\mu\nu}^d = \left\{ g_{\mu\nu}^d, \Omega_{\mu\nu}^d, T_{\mu\nu}^d \right\}$.

## 2.2 Zero temperature and noninteracting limit

In this section, we justify the definitions of $\left\{g_{\mu\nu}^d, \Omega_{\mu\nu}^d, T_{\mu\nu}^d\right\}$ in Sec. 2.1 by showing that they recover the $\left\{g_{\mu\nu}, \Omega_{\mu\nu}, T_{\mu\nu}\right\}$ in the noninteracting and zero temperature limit given by Eqs. (2) and (2). We first write the fully antisymmetric filled band state in the second quantized form (ignoring the momentum index $\mathbf{k}$) $|\psi^{\text{val}}\rangle = \sum_{n\in v} c_n^\dagger |0\rangle$ and introduce the projection operators for the filled band $Q_- = \sum_{n\in v} |n\rangle\langle n|$ and empty band $Q_+ = \sum_{m\in c} |m\rangle\langle m|$. The inner product of the derivatives of $|\psi^{\text{val}}\rangle$ is

$$\langle\partial_\mu\psi^{\text{val}}|\partial_\nu\psi^{\text{val}}\rangle = \left(\sum_{n\in v}\langle\partial_\mu n|n\rangle\right)\left(\sum_{n\in v}\langle n|\partial_\nu n\rangle\right) + \sum_{n\in v}\langle\partial_\mu n|Q_+|\partial_\nu n\rangle, \quad (10)$$

which gives the noninteracting Berry curvature and quantum metric

$$\Omega_{\mu\nu} = \sum_{n\in v}\left[i\langle\partial_\mu n|\partial_\nu n\rangle - i\langle\partial_\nu n|\partial_\mu n\rangle\right],$$

$$g_{\mu\nu} = \frac{1}{2}\sum_{n\in v}\left[\langle\partial_\mu n|Q_+|\partial_\nu n\rangle + \langle\partial_\nu n|Q_+|\partial_\mu n\rangle\right]. \quad (11)$$

On the other hand, in the noninteracting limit of the Green's function $G \to G^{(0)}$, the dynamic fidelity susceptibility is given by

$$\chi_{\mu\nu}^{(0)}(\mathbf{k}, i\omega) = \sum_{nm}\mathcal{A}_\mu^{nm}\left[\mathcal{A}_\nu^{nm}\right]^\dagger \frac{1}{\beta}\sum_{ip}G_n^{(0)}(\mathbf{k}, ip)G_m^{(0)}(\mathbf{k}, i\omega + ip). \quad (12)$$

The frequency sum gives the usual Lindhard function, so the unperturbed real frequency susceptibility is (suppressing $\mathbf{k}$ index for simplicity)

$$\chi_{\mu\nu}^{(0)}(\omega) = \sum_{nm}\langle\partial_\mu n|m\rangle\langle m|\partial_\nu n\rangle \frac{f(E_n) - f(E_m)}{\omega + E_n - E_m + i\eta}. \quad (13)$$

Let us first consider zero temperature limit such that the Fermi functions are step functions, which demand $E_n$ must belong to the valence bands $n \in v$ and $E_m$ the conduction bands $m \in c$. Symmetrizing the imaginary part yields

$$-\frac{1}{2\pi}\text{Im}\left[\chi_{\mu\nu}^{(0)}(\omega) + \chi_{\nu\mu}^{(0)}(\omega)\right]_{T=0} = \sum_{n\in v, m\in c}\frac{1}{2}\left[\langle\partial_\mu n|m\rangle\langle m|\partial_\nu n\rangle + (\mu \leftrightarrow \nu)\right]\delta(\omega + E_n - E_m). \quad (14)$$

After an integration over frequency, one obtains the zero temperature and noninteracting limit of the dressed quantum metric $g_{\mu\nu}^d(\mathbf{k})|_{H'=0, T=0} = g_{\mu\nu}(\mathbf{k})$, which recovers that of the filled band Bloch state in Eq. (11).

To see the Berry curvature, one may consider the combination $i\chi_{\mu\nu}^{(0)}(\omega)|_{T=0} - i\chi_{\nu\mu}^{(0)}(\omega)|_{T=0}$ and integrate it over frequency

$$i\int d\omega\, \chi_{\mu\nu}^{(0)}(\omega)|_{T=0} - i\int d\omega\, \chi_{\nu\mu}^{(0)}(\omega)|_{T=0} =$$

$$= \sum_{n\in v, m\in c}\left[i\langle\partial_\mu n|m\rangle\langle m|\partial_\nu n\rangle - i\langle\partial_\nu n|m\rangle\langle m|\partial_\mu n\rangle\right]\int\frac{d\omega}{\omega + E_n - E_m}$$

$$+ \pi\sum_{n\in v, m\in c}\left[\langle\partial_\mu n|m\rangle\langle m|\partial_\nu n\rangle - \langle\partial_\nu n|m\rangle\langle m|\partial_\mu n\rangle\right]. \quad (15)$$

The second line above is purely real and the third line purely imaginary, and hence

$$\frac{i}{\pi}\text{Re}\left[\int d\omega\,\chi^{(0)}_{\mu\nu}(\omega)|_{T=0} - \int d\omega\,\chi^{(0)}_{\nu\mu}(\omega)|_{T=0}\right] =$$
$$= \sum_{n\in\nu}\left[\langle\partial_\mu n|(I-Q_-)|\partial_\nu n\rangle - \langle\partial_\nu n|(I-Q_-)|\partial_\mu n\rangle\right]$$
$$= \sum_{n\in\nu}\left[\langle\partial_\mu n|\partial_\nu n\rangle - \langle\partial_\nu n|\partial_\mu n\rangle\right]. \tag{16}$$

Thus the dressed Berry curvature in the zero temperature and noninteracting limit $\Omega^d_{\mu\nu}(\mathbf{k})|_{H'=0,T=0} = \Omega_{\mu\nu}(\mathbf{k})$ recovers the noninteracting Berry curvature in Eq. (11).

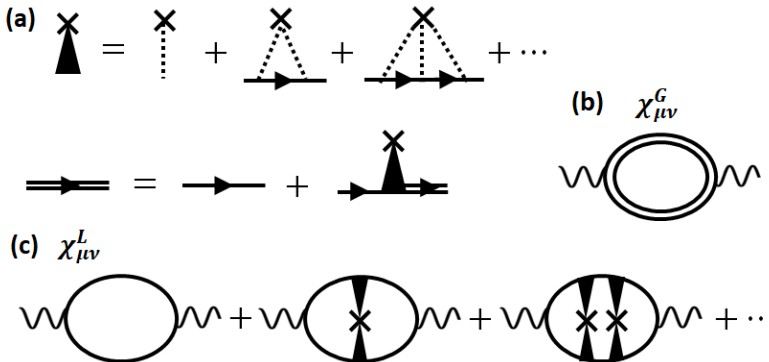

Figure 1: (a) Feynman diagrams for the self-energy $\Sigma$ caused by impurity scattering, and the full Green's function solved by Dyson's equation $G = G^{(0)} + G^{(0)}\Sigma G$. (b) The susceptibility $\chi^G_{\mu\nu}$ calculated from the full Green's function, and (c) $\chi^L_{\mu\nu}$ calculated in the ladder diagram approximation.

## 2.3 Perturbative calculation of dressed Berry curvature and quantum metric

In the presence of interactions $H'$, there are various approximations that can be used to calculate the susceptibility. For concreteness, in present work we discuss two most frequently used approximations, which will be applied to a concrete example in Sec. 2.5. The first uses the full Green's function calculated from the Dyson's equation $G = G^{(0)} + G^{(0)}\Sigma G$ in the polarization operator, as indicated in Fig. 1 (a) and (b) using impurity scattering as an example, which yields

$$\chi^G_{\mu\nu}(\mathbf{k}, i\omega) = \sum_{nm}\mathcal{A}^{nm}_\mu\left[\mathcal{A}^{nm}_\nu\right]^\dagger\frac{1}{\beta}\sum_{ip}G_n(\mathbf{k}, ip)G_m(\mathbf{k}, i\omega + ip). \tag{17}$$

One may use the single-particle spectral function for the full Green's function $A(\mathbf{k}, \omega) = -\text{Im}\,G(\mathbf{k}, \omega)/\pi$ to rewrite Eq. (17), yielding [37]

$$g^G_{\mu\nu}(\mathbf{k}, \omega) = \sum_{nm}\frac{1}{2}\left\{\mathcal{A}^{nm}_\mu\left[\mathcal{A}^{nm}_\nu\right]^\dagger + \mathcal{A}^{nm}_\nu\left[\mathcal{A}^{nm}_\mu\right]^\dagger\right\}$$
$$\times\int d\varepsilon\,A_n(\mathbf{k}, \varepsilon)A_m(\mathbf{k}, \varepsilon + \omega)\left[f(\varepsilon) - f(\varepsilon + \omega)\right], \tag{18}$$

and likewisely for $\Omega^G_{\mu\nu}(\mathbf{k}, \omega)$, where $f(\epsilon)$ is the Fermi distribution that determines the filling at finite temperature. One sees that the self-energy broadens the spectral function $A(\mathbf{k}, \omega)$, and

subsequently broadens $g_{\mu\nu}^G(\mathbf{k},\omega)$ and $\Omega_{\mu\nu}^G(\mathbf{k},\omega)$ from the $\delta$-functions peaking at $\omega = E_{m\mathbf{k}} - E_{n\mathbf{k}}$, as explained in Appendix A using a toy model with artificial broadening.

Another frequently used approximation are the ladder diagrams in Fig. 1 (c) that correspond to

$$\chi_{\mu\nu}^L(\mathbf{k}, i\omega) = \sum_{nm} \mathcal{A}_\mu^{nm}(\mathbf{k}) \frac{1}{\beta} \sum_{ip} G_n^{(0)}(\mathbf{k}, ip) G_m^{(0)}(\mathbf{k}, ip+i\omega) \Gamma_\nu^{nm}(\mathbf{k}, ip, ip+i\omega), \qquad (19)$$

where the vertex function $\Gamma_\nu^{nm}$ acts like a dressed non-Abelian gauge field. We will use the intraband impurity scattering as an example, in which $\Gamma_\nu^{nm}$ satisfies the Bethe-Salpeter equation (BSE) [37]

$$\Gamma_\nu^{nm}(\mathbf{k}, ip, ip+i\omega) = \left[\mathcal{A}_\nu^{nm}(\mathbf{k})\right]^\dagger + \sum_{\mathbf{k}'} W_{\mathbf{k}\mathbf{k}'}^{nm}(i\omega) G_n^{(0)}(\mathbf{k}', ip) G_m^{(0)}(\mathbf{k}', ip+i\omega) \Gamma_\nu^{nm}(\mathbf{k}', ip, ip+i\omega), \quad (20)$$

where $W_{\mathbf{k}\mathbf{k}'}^{nm}(i\omega)$ is the impurity scattering vertex. The $G^{(0)}$ may be replaced by the full Green's function $G$ in more sophisticated calculations.

## 2.4 Measurements by exciton or infrared absorption rate

The oscillating electric field is expected to cause particle-hole excitations even at finite temperature, which may be detected by exciton absorption in semiconductors and infrared absorption in superconductors. In time-dependent perturbation theory with the perturbation $\delta h(\mathbf{k}, t) = -iqE_0 e^{-i\omega t} \partial_\mu$, one can immediately identify the exciton absorption rate $R(\mathbf{k}, \omega)$ obtained from the Fermi golden rule with our quantum metric spectral function [36] (in standard unit)

$$R(\mathbf{k}, \omega) = 2\pi \left(\frac{qE_0}{\hbar}\right)^2 g_{\mu\mu}^d(\mathbf{k}, \omega). \qquad (21)$$

The off-diagonal components, for instance $g_{xy}^d(\mathbf{k}, \omega)$ defined in the $xy$-plane, can be extracted by considering two different measurement protocols [17] that applied the same force strength $qE_0$ in the two directions but with a phase difference $\pm 1$

$$\delta h^{(\pm)} = \left(U_x^\dagger \pm U_y^\dagger\right) qE_0 e^{-i\omega t}, \qquad (22)$$

which induces the polarization

$$\langle U_x(\mathbf{k}, t) \pm U_y(\mathbf{k}, t) \rangle = \chi^{(\pm)}(\mathbf{k}, t) qE_0 e^{-i\omega t}, \qquad (23)$$

where $\chi^{(\pm)} = \chi_{xx} \pm \chi_{xy} \pm \chi_{yx} + \chi_{yy}$, and hence subtracting the two absorption rates yields

$$R^{(+)}(\mathbf{k}, \omega) - R^{(-)}(\mathbf{k}, \omega) = 2\pi \left(\frac{qE_0}{\hbar}\right)^2 4g_{xy}^d(\mathbf{k}, \omega). \qquad (24)$$

After $g_{\mu\nu}^d(\mathbf{k})$ is measured, various differential geometric quantities that characterize the momentum space manifold like Ricci scalar, Riemann tensor, and geodesics (in the noninteracting limit, it is the trajectory along which the Bloch state rotates the least) can be extracted according to their usual definitions in terms of $g_{\mu\nu}^d(\mathbf{k})$. Likewisely, the Berry curvature can be extracted by applying the same force in the two directions but with a phase difference $\pm i$ [34]

$$\delta h^{c1,c2} = \left(U_x^\dagger \pm iU_y^\dagger\right) qE_0 e^{-i\omega t}, \qquad (25)$$

which are precisely the two circular polarizations, causing the polarization

$$\langle U_x(\mathbf{k}, t) \mp iU_y(\mathbf{k}, t) \rangle = \chi^{c1,c2}(\mathbf{k}, t) qE_0 e^{-i\omega t}, \qquad (26)$$

where $\chi^{c1,c2} = \chi_{xx} \pm i\chi_{xy} \mp i\chi_{yx} + \chi_{yy}$. Subtracting the absorption rates of the two protocols, i.e., a circular dichroism measurement, yields

$$R^{c1}(\mathbf{k}, \omega) - R^{c2}(\mathbf{k}, \omega) = 2\pi \left( \frac{qE_0}{\hbar} \right)^2 2\Omega_{xy}^d(\mathbf{k}, \omega), \tag{27}$$

which gives the Berry curvature spectral function.

Experimental techniques that can resolve the momentum and frequency dependence of exciton absorption rate can directly measure $g_{\mu\nu}^d(\mathbf{k}, \omega)$ and $\Omega_{\mu\nu}^d(\mathbf{k}, \omega)$. Note that the usual exciton absorption experiment measures the spectral function integrated over momentum $\mathbf{k}$ and plotted as a function of $\omega$ [36,43–45], but our proposal requires to integrate it over $\omega$ and plot it as a function of $\mathbf{k}$. To serve this purpose, we anticipate that the most promising technique may be time-resolved and angle-resolved photoemission spectroscopy (trARPES) [46–51]. In this technique, the change of particle number in all the valence bands $\Delta n_\nu(\mathbf{k}, \omega, t)$ and in all the conduction bands $\Delta n_c(\mathbf{k}, \omega, t)$ at $\mathbf{k}$ after the electrostatic force $B_0 e^{-i\omega t} = qE_0 e^{-i\omega t}$ polarized along $\mu$ has been applied for time $t$ is

$$\Delta n_c(\mathbf{k}, \omega, t) = -\Delta n_\nu(\mathbf{k}, \omega, t) = R(\mathbf{k}, \omega) t$$

$$= N_- - z(\mathbf{k}) \int_{-\infty}^0 d\varepsilon_1 \sum_{n \in \nu} A_n(\mathbf{k}, \varepsilon_1) f^*(\varepsilon_1, \omega, t) \tag{28}$$

$$= z(\mathbf{k}) \int_0^\infty d\varepsilon_1 \sum_{m \in c} A_c(\mathbf{k}, \varepsilon_1) f^*(\varepsilon_1, \omega, t),$$

where $f^*(\varepsilon, \omega, t)$ represents a nonequilibrium Fermi distribution function that evolves with time, and the phenomenological fitting parameter $z(\mathbf{k})$ can be used to adjust the experimentally measured $A_n(\mathbf{k}, \varepsilon)$ until the spectral sum rule at equilibrium $N_- = z(\mathbf{k}) \int_{-\infty}^0 d\varepsilon_1 \sum_{n \in \nu} A_n(\mathbf{k}, \varepsilon_1) f(\varepsilon_1)$ is satisfied. Equation (37) provides a measurement protocol for $g_{\mu\nu}^d(\mathbf{k}, \omega)$ and $\Omega_{\mu\nu}^d(\mathbf{k}, \omega)$ in the proposed trARPES experiment, in which one measures the lost of particle number in the valence bands or the gain of particle number in the conduction bands after the electric field $E^\mu$ with frequency $\omega$ has been applied for time $t$.

## 2.5 Disordered Chern insulator in a continuum

We proceed to use Chern insulator in a continuum with impurity scattering as a concrete example. This example is chosen for multiple reasons. Firstly, analytical results for the self-energy can be given, from which the broadening and shift of single-particle spectral function and how they subsequently affect the Berry curvature spectral function and quantum metric spectral function can be clearly demonstrated. Secondly, the noninteracting Chern insulator has topological order, and therefore how the disorder affects the topological and quantum geometrical property of the system can be unambiguously understood. Thirdly, this simple model serves as a good example to demonstrate how the band gap protects the topological and quantum geometrical properties against many-body interactions, which must be understood before other factors, such as realistic band structures, spin or orbital degrees of freedom, etc., should be investigated. The single particle Hamiltonian of this model is expanded by the Pauli matrices $h(\mathbf{k}) = \mathbf{d}(\mathbf{k}) \cdot \boldsymbol{\sigma}$, with $d_1 = vk_x$, $d_2 = vk_y$, and $d_3 = M$, where $v = 1$ is the Fermi velocity and $M$ represents the band gap. The model contains only one filled band and one empty band, and the modulus of momentum is restricted to $0 \le k \le \pi/a$ such that the integration in the self-energy is finite, where $a = 1$ represents a lattice constant. In the noninteracting and zero temperature limit, the square root of the determinant of the quantum metric is equal to half of the module of the Berry curvature [11,13–15,52]

$$\sqrt{\det g_{\mu\nu}} = |\Omega_{xy}|/2, \tag{29}$$

a relation that is a special case of the so-called metric-curvature correspondence [33] that has been derived from a universal topological invariant [53]. Whether such a relation still holds in the presence of interactions would be a good indication of whether the quantum geometric properties remain unchanged. The Chern insulator in the presence of electron-electron and electron-phonon interactions has been considered previously [54, 55], but we will consider the intraband impurity scattering that does not transfer electrons between the two bands. Details of the calculation is given in Appendix B, including the argument to ignore the ladder diagrams, so we only focus on the $g_{\mu\nu}^G(\mathbf{k}, \omega)$ and $\Omega_{\mu\nu}^G(\mathbf{k}, \omega)$ defined from Eq. (18).

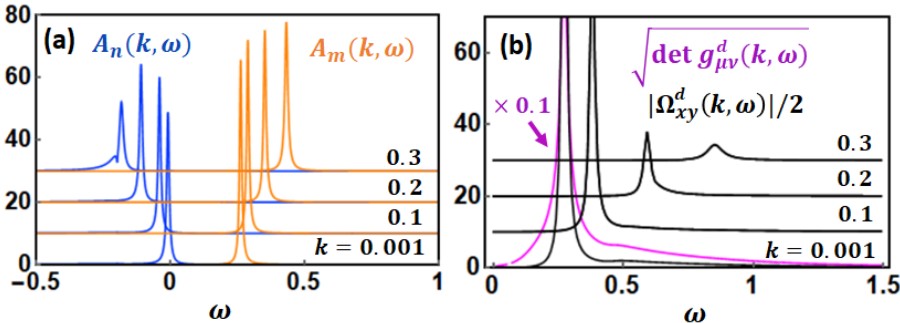

Figure 2: (a) Single-particle spectral function of the Chern insulator with impurity density $n_i = 0.1$ and impurity potential $V = 1$, plotted for several diagonal momenta $k_x = k_y = k$. Each line is shifted upward for the sake of presentation. The chemical potential is set at $\mu = 0.13$ and temperature at $k_B T = 0.03$. (b) The Berry curvature spectral function $|\Omega_{xy}^d(k, \omega)|/2$ and quantum metric spectral function $\sqrt{\det g_{\mu\nu}^d(k, \omega)}$, which coincide at large momenta, signifying the metric-curvature correspondence, but deviate at small momenta due to the reduced band gap.

Figure 2 (a) shows the single-particle spectral function of this model at different $\mathbf{k}$, where the impurity scattering shifts and broadens the quasiparticle peak as expected, and the band gap can be identified from the peak positions. The module of Berry curvature spectral function $|\Omega_{xy}^d(\mathbf{k}, \omega)|/2$ and the square root of the determinant of quantum metric spectral function $\sqrt{\det g_{\mu\nu}^d(\mathbf{k}, \omega)}$ shown in Fig. 2 (b) peak at the band gap, reminisce the feature of exciton absorption rates. At large momentum and large band gap, the coincidence of the two spectral functions indicate that Eq. (29) is satisfied, signifying the band gap protects the geometric properties against the interaction. However, at small momentum, the two spectral functions deviate significantly, suggesting that interactions can alter the quantum geometric properties in regions with a small band gap, which is in accordance with our phenomenological explanation using an artificial broadening given in the supplemental material.

## 3 Conclusions

In summary, we have presented a formalism of quantum metric and Berry curvature for realistic gapped materials at finite temperature and subject to many-body interactions. Our formalism is based on the linear response theory of charge polarization induced by polarized electric field, which recognizes the real frequency charge polarization susceptibility as the spectral functions of quantum metric and Berry curvature. The spectral functions are also the exciton or infrared absorption rate caused by the polarized electric field, suggesting a concrete proto-

col to measure these quantities even at finite temperature and in the presence of many-body interactions. The spectral functions integrated over frequency give the dressed Berry curvature and quantum metric at momentum $\mathbf{k}$, and hence experimental techniques that can measure exciton absorption rate with a momentum resolution, such as the loss of valence band spectral weight measured by trARPES, can directly detect these quantities.

The perturbative calculation of the spectral functions is analogous to that in the theory of exciton absorption rate in semiconductors induced by minimal coupling. Using disordered Chern insulator as an example, we reveal that the spectral functions are significantly broadened by interactions, as expected. However, within the full Green's function approximation and the ladder diagrams approximation, our results suggest that the quantum geometric properties of the Chern insulator is protected by the energy gap against interactions, in the sense that the metric-curvature correspondence between Berry curvature and quantum metric remains unchanged if the energy gap is larger than the strength of the impurity scattering. Finally, as our formalism is broadly applicable to any semiconductors, superconductors, and topological insulators, we anticipate that the influence of temperature and interactions on the quantum geometric properties of a variety of gapped material can be investigated ubiquitously within our linear response theory. On the other hand, we also anticipate that when combining our linear response theory with the realistic band structures obtained from first-principle calculations, a lot of technical details may arise, such as Wannierization [38], which are important issues that await to be explored.

We thank exclusively A. F. Kemper for the discussion about various aspects related to pump-probe experiments. W. C. is financially supported by the productivity in research fellowship from CNPq.

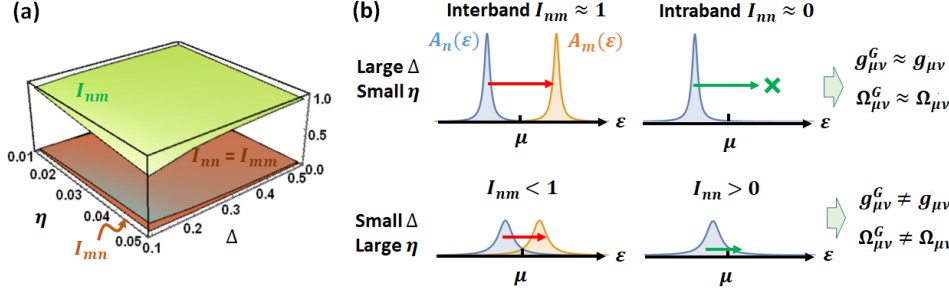

Figure 3: (a) The integrals $\{I_{nn}, I_{nm}, I_{mn}, I_{mm}\}$ that enter the expression of quantum metric in Eq. (31) for our two-band toy model, plotted as a function of the band gap $\Delta$ and artificial broadening $\eta$. (b) Schematics of the interband and intraband transition processes at weak (top) and strong (bottom) interactions, and why in the later case the dressed quantum metric and Berry curvature deviate from their noninteracting values.

# A  Two-band toy model with artificial broadening

In the section, we use a two-band toy model to schematically demonstrate how the broadening of single-particle spectral function by interaction causes the Berry curvature and quantum metric to deviate from their noninteracting values. Consider a model that contains only one filled band state $|n\rangle$ with energy $-\Delta$ and one empty band state $|n\rangle$ with energy $+\Delta$, which give some form of $\mathcal{A}_\mu^{nm}$ that is not important at this stage (all of these are functions of momentum $\mathbf{k}$, but we omit this index for simplicity). The spectral functions are assumed to take the

Lorentzian shape with an artificial broadening $\eta$ that comes from some source of scattering

$$A_n(\varepsilon) = \frac{\eta/\pi}{(\varepsilon+\Delta)^2+\eta^2} , A_m(\varepsilon) = \frac{\eta/\pi}{(\varepsilon-\Delta)^2+\eta^2} , \tag{30}$$

and we restrict the discussion to zero temperature such that the Fermi functions are step functions $f(\varepsilon) = \theta(-\varepsilon)$. As a result, the quantum metric $g_{\mu\mu}^G$ calculated using spectral representation, given in Eq. (12) of the main text, contains four terms ($\sum_{n'}$ and $\sum_{m'}$ both sum the two bands)

$$g_{\mu\mu}^G = \sum_{n'm'} A_\mu^{n'm'} \left[A_\mu^{n'm'}\right]^\dagger \times I_{n'm'} ,$$

$$I_{n'm'} \equiv \int_0^\infty d\omega \int_{-\omega}^0 d\varepsilon A_{n'}(\varepsilon) A_{m'}(\varepsilon+\omega) . \tag{31}$$

Out of the four integrations $\{I_{nn}, I_{nm}, I_{mn}, I_{mm}\}$, the $\{I_{nm}, I_{mn}\}$ represent the interband and $\{I_{nn}, I_{mm}\}$ the intraband transitions. In the noninteracting limit $\lim_{\eta\to0} A_n(\varepsilon) = \delta(\varepsilon+\Delta)$ and $\lim_{\eta\to0} A_m(\varepsilon) = \delta(\varepsilon-\Delta)$, only the $I_{nm} = 1$ gives unity and all others are zero $I_{mn} = I_{nn} = I_{mm} = 0$, so the noninteracting quantum metric is simply $g_{\mu\mu}^G = A_\mu^{nm}\left[A_\mu^{nm}\right]^\dagger = \langle\partial_\mu n|m\rangle\langle m|\partial_\mu n\rangle = g_{\mu\mu}$.

In the presence of interaction $\eta \neq 0$, how much $I_{nm}$ deviates from unity and how much $\{I_{nn}, I_{mn}, I_{mm}\}$ deviate from zero would give us a sense of how much $g_{\mu\mu}^G$ deviates from $g_{\mu\mu}$, which obviously depends on the strength of interaction $\eta$ and the band gap $\Delta$. Figure 3 (a) shows the numerical result of $\{I_{nn}, I_{nm}, I_{mn}, I_{mm}\}$ for this toy model. At large gap $\Delta$ and small broadening $\eta$, the spectral functions $A_n(\varepsilon)$ and $A_m(\varepsilon)$ are well separated peaks whose shapes are close to $\delta$-functions, leading to the interband transition amplitude $I_{nm} \approx 1$. Because $A_n(\varepsilon)$ has a negligible weight above chemical potential $\varepsilon > \mu$, the intraband transition amplitude is practically zero $I_{nn} \approx 1$. As a result, the dressed quantum metric and Berry curvature roughly preserve their noninteracting values $g_{\mu\nu}^G \approx g_{\mu\nu}$ and $\Omega_{\mu\nu}^G \approx \Omega_{\mu\nu}$. In contrast, at small gap $\Delta$ and large broadening $\eta$, signifying strong interactions, the spectral functions $A_n(\varepsilon)$ and $A_m(\varepsilon)$ overlap significantly and each has notable weight above or below the chemical potential, causing $I_{nm} < 1$ and $I_{nn} > 0$. After multiplying by the matrix elements of non-Abelian gauge fields, these deviations cause $g_{\mu\nu}^G$ and $\Omega_{\mu\nu}^G$ to differ from their noninteracting values. Although this result is in accordance with the expectation that the band gap protects the quantum geometric properties against any source of interactions, it should be noted that even for broadening $\eta$ as small as 20% of the gap $\Delta$ there is already a notable change of $I_{nm}$ and $I_{nn}$. For instance, at $\Delta = 0.2$ and $\eta = 0.04$, where the two Lorentzian peaks $A_n(\varepsilon)$ and $A_m(\varepsilon)$ appeared to be very apart, the interband transition is already reduced to $I_{nm} \approx 0.878$ and the intraband transition increased to $I_{nn} \approx 0.059$.

# B  Detail of the Chern insulator with impurity scattering

We now detail the susceptibility $\chi_{\mu\nu}(\mathbf{k}, \omega)$ for Chern insulator in a continuum with impurities. Parametrizing the $2 \times 2$ Dirac Hamiltonian by

$$H = \mathbf{d} \cdot \boldsymbol{\sigma} = d_1\sigma_1 + d_2\sigma_2 + d_3\sigma_3 , \tag{32}$$

the components are given by $d_1 = vk_x$, $d_2 = vk_y$, and $d_3 = M$. Denoting $d = \sqrt{d_1^2 + d_2^2 + d_3^2}$, the filled band state $|n\mathbf{k}\rangle$ with energy $E_{n\mathbf{k}} = -d$ and the empty band state $|m\mathbf{k}\rangle$ with energy

$E_{m\mathbf{k}} = d$ are given by

$$|n, m\mathbf{k}\rangle = \frac{1}{\sqrt{2d(d \mp d_3)}} \begin{pmatrix} d_3 \mp d \\ d_1 + id_2 \end{pmatrix}, \tag{33}$$

where the upper sign is for $|n\mathbf{k}\rangle$ and the lower sign $|m\mathbf{k}\rangle$. In this gauge, the non-Abelian gauge field takes the form

$$\mathcal{A}_\mu^{nn} = \langle n|i\partial_\mu|n\rangle = \frac{d_2 \partial_\mu d_1 - d_1 \partial_\mu d_2}{2d(d - d_3)},$$

$$\mathcal{A}_\mu^{mm} = \langle m|i\partial_\mu|m\rangle = \frac{d_2 \partial_\mu d_1 - d_1 \partial_\mu d_2}{2d(d + d_3)}, \tag{34}$$

$$\mathcal{A}_\mu^{nm} = \langle n|i\partial_\mu|m\rangle = \frac{d_2 \partial_\mu d_1 - d_1 \partial_\mu d_2 - id \partial_\mu d_3 + id_3 \partial_\mu d}{2d\sqrt{d_1^2 + d_2^2}} = \left(\mathcal{A}_\mu^{mn}\right)^*.$$

In the $xy$-plane of the continuous Chern insulator, they are

$$\mathcal{A}_x^{nn} = \frac{v^2 k_y}{2d(d - M)}, \qquad \mathcal{A}_x^{mm} = \frac{v^2 k_y}{2d(d + M)}, \qquad \mathcal{A}_y^{nn} = -\frac{v^2 k_x}{2d(d - M)},$$

$$\mathcal{A}_y^{mm} = -\frac{v^2 k_x}{2d(d + M)}, \qquad \mathcal{A}_x^{nm} = \frac{v^2 k_y + iM v^2 k_x/d}{2dvk} = \left(\mathcal{A}_x^{mn}\right)^*, \tag{35}$$

$$\mathcal{A}_y^{nm} = \frac{-v^2 k_x + iM v^2 k_y/d}{2dvk} = \left(\mathcal{A}_y^{mn}\right)^*.$$

The bare retarded Green's function $G_n^{(0)}(\mathbf{k}, \omega) = G_n^{(0)}(k, \omega)$ does not depend on the azimuthal angle $\varphi$ but only the module of the momentum $k$. Assuming only intraband scattering, the impurity potential $V \times I_{2\times 2}$ gives the matrix elements

$$V_{\mathbf{k}\mathbf{k}'}^n = \langle n\mathbf{k}'|V|n\mathbf{k}\rangle = \frac{V}{2d(d - d_3)}\left[(d_3 - d)^2 + (d_1^2 + d_2^2)e^{i(\varphi - \varphi')}\right],$$

$$V_{\mathbf{k}\mathbf{k}'}^m = \langle m\mathbf{k}'|V|m\mathbf{k}\rangle = \frac{V}{2d(d + d_3)}\left[(d_3 + d)^2 + (d_1^2 + d_2^2)e^{i(\varphi - \varphi')}\right]. \tag{36}$$

The $T$-matrix of impurity scattering satisfies the self-consistent equation

$$\begin{aligned}
T_{\mathbf{k}\mathbf{k}'}^{n/m}(\omega) &= V_{\mathbf{k}\mathbf{k}'}^{n/m} + \int_0^{2\pi} \frac{d\varphi_1}{2\pi} \int_0^{\pi/a} \frac{k_1 \, dk_1}{2\pi/a^2} V_{\mathbf{k}\mathbf{k}_1}^{n/m} T_{\mathbf{k}_1\mathbf{k}'}^{n/m}(\omega) G_{n/m}^{(0)}(k_1, \omega) \\
&= \frac{V}{2}\left(\frac{d \pm d_3}{d}\right)\left[1 + b e^{i(\varphi - \varphi')}\right] \\
&\quad + \left[\frac{V}{2}\left(\frac{d \pm d_3}{d}\right)\right]^2 \left[1 + b^2 e^{i(\varphi - \varphi')}\right] \int_0^{\pi/a} \frac{k_1 \, dk_1}{2\pi/a^2} G_{n/m}^{(0)}(k_1, \omega) + \dots,
\end{aligned} \tag{37}$$

where $b = (d_1^2 + d_2^2)/(d \pm d_3)^2$. The radial integration of retarded Green's function can be performed analytically by

$$\int_0^{\pi/a} \frac{k_1 \, dk_1}{2\pi/a^2} G_{n/m}^{(0)}(k_1, \omega) = \int_0^{\pi/a} \frac{k_1 \, dk_1}{2\pi/a^2}\left[\frac{1}{\omega \pm d} - \frac{i\eta}{(\omega \pm d)^2 + \eta^2}\right], \tag{38}$$

where $\eta$ is an artificial broadening, whose real and imaginary parts are

$$\mathrm{Re}^{n/m} = \frac{a^2}{2\pi v^2}\left\{\pm\left(\tilde{M} - |M|\right) - \omega \ln\left|\frac{\omega \pm \tilde{M}}{\omega \pm |M|}\right|\right\},$$

$$\mathrm{Im}^{n/m} = \frac{\eta a^2}{2\pi v^2}\left\{\pm\frac{\omega}{\eta}\left[\arctan\frac{\tilde{M} \pm \omega}{\eta} - \arctan\frac{|M| \pm \omega}{\eta}\right] - \frac{1}{2}\ln\left|\frac{(\tilde{M} \pm \omega)^2 + \eta^2}{(|M| \pm \omega)^2 + \eta^2}\right|\right\}. \tag{39}$$

After an impurity averaging, the self-energy is given by impurity density multiplied by the $T$-matrix at the same momentum index $\Sigma_{n/m}(\mathbf{k}, \omega) = n_i T_{\mathbf{kk}}^{n/m}(\omega)$, which can then be used to calculate the spectral function $A_n(\mathbf{k}, \omega) = -\mathrm{Im}\, G_n(\mathbf{k}, \omega)/\pi$, yielding

$$A_n(\mathbf{k}, \omega) = -\frac{1}{\pi} \frac{\mathrm{Im}\Sigma(\mathbf{k}, \omega)}{(\omega - E_{n\mathbf{k}} - \mathrm{Re}\Sigma(\mathbf{k}, \omega))^2 + \mathrm{Im}\Sigma(\mathbf{k}, \omega)^2}, \tag{40}$$

and subsequently the susceptibility $\chi_{\mu\nu}^G$ that uses the full Green's function.

For the ladder diagrams of the susceptibility, the Matsubara four-fermion vertex that enters the Feynman diagrams is given by the $T$-matrix

$$W_{\mathbf{kk}'}^{nm}(i\omega) = n_i T_{\mathbf{kk}'}^n(i\omega) T_{\mathbf{kk}'}^m(i\omega), \tag{41}$$

which does not transfer frequency between the filled band propagator and the empty band propagator. As a result, the vertex function $\Gamma_\nu^{nm}$ in the ladder diagrams satisfies

$$
\begin{aligned}
\Gamma_\nu^{nm}(\mathbf{k}, ip, ip + i\omega) &= \left[\mathcal{A}_\nu^{nm}(\mathbf{k})\right]^\dagger \\
&+ \sum_{\mathbf{k}'} W_{\mathbf{kk}'}^{nm}(i\omega) G_n^{(0)}(\mathbf{k}', ip) G_m^{(0)}(\mathbf{k}', ip + i\omega) \Gamma_\nu^{nm}(\mathbf{k}', ip, ip + i\omega) \\
&= \left[\mathcal{A}_\nu^{nm}(\mathbf{k})\right]^\dagger \left\{ 1 + \sum_{\mathbf{k}'} W_{\mathbf{kk}'}^{nm}(i\omega) G_n^{(0)}(\mathbf{k}', ip) G_m^{(0)}(\mathbf{k}', ip + i\omega) + ... \right\}.
\end{aligned} \tag{42}
$$

The first term in the last line gives the bare susceptibility $\chi_{\mu\nu}^{(0)}$. The second order term, after inserting it back to the expression of ladder diagrams, will contribute to a frequency sum of four propagators

$$-\frac{1}{\beta} \sum_{ip} S(i\omega, ip) = \frac{1}{\beta} \sum_{ip} \frac{1}{ip - E_{n\mathbf{k}}} \frac{1}{ip + i\omega - E_{m\mathbf{k}}} \frac{1}{ip - E_{n\mathbf{k}'}} \frac{1}{ip + i\omega - E_{m\mathbf{k}'}}. \tag{43}$$

Performing the frequency sum and subsequently an analytical continuation $i\omega \to \omega + i\eta$, and then taking the imaginary part to get the spectral function, this second order term gives

$$
\begin{aligned}
-\frac{1}{\pi}\mathrm{Im}\left\{ -\frac{1}{\beta} \sum_{ip} S(i\omega, ip) \right\}_{i\omega \to \omega + i\eta} &= \left[ \frac{\delta(\omega + E_{n\mathbf{k}} - E_{m\mathbf{k}})}{\omega + E_{n\mathbf{k}} - E_{m\mathbf{k}'}} + \frac{\delta(\omega + E_{n\mathbf{k}} - E_{m\mathbf{k}'})}{\omega + E_{n\mathbf{k}} - E_{m\mathbf{k}}} \right] \frac{f(E_{n\mathbf{k}})}{E_{n\mathbf{k}} - E_{n\mathbf{k}'}} \\
&+ \left[ \frac{\delta(\omega + E_{n\mathbf{k}} - E_{m\mathbf{k}})}{\omega + E_{n\mathbf{k}'} - E_{m\mathbf{k}}} + \frac{\delta(\omega + E_{n\mathbf{k}'} - E_{m\mathbf{k}})}{\omega + E_{n\mathbf{k}} - E_{m\mathbf{k}}} \right] \frac{f(E_{m\mathbf{k}})}{E_{m\mathbf{k}} - E_{m\mathbf{k}'}} \\
&+ \left[ \frac{\delta(\omega + E_{n\mathbf{k}'} - E_{m\mathbf{k}'})}{\omega + E_{n\mathbf{k}'} - E_{m\mathbf{k}}} + \frac{\delta(\omega + E_{n\mathbf{k}'} - E_{m\mathbf{k}})}{\omega + E_{n\mathbf{k}'} - E_{m\mathbf{k}'}} \right] \frac{f(E_{n\mathbf{k}'})}{E_{n\mathbf{k}'} - E_{n\mathbf{k}}} \\
&+ \left[ \frac{\delta(\omega + E_{n\mathbf{k}'} - E_{m\mathbf{k}'})}{\omega + E_{n\mathbf{k}} - E_{m\mathbf{k}'}} + \frac{\delta(\omega + E_{n\mathbf{k}} - E_{m\mathbf{k}'})}{\omega + E_{n\mathbf{k}'} - E_{m\mathbf{k}'}} \right] \frac{f(E_{m\mathbf{k}'})}{E_{m\mathbf{k}'} - E_{m\mathbf{k}}},
\end{aligned} \tag{44}
$$

which vanishes after a frequency integration

$$-\frac{1}{\pi} \int d\omega \, \mathrm{Im}\left\{ -\frac{1}{\beta} \sum_{ip} S(i\omega, ip) \right\}_{i\omega \to \omega + i\eta} = 0. \tag{45}$$

We conclude that this second order term does not contribute to the quantum metric or Berry curvature. The next order in the ladder diagrams is proportional to the impurity density square $n_i^2$, which may be ignored.

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
