# Peer review of "Measurement of interaction-dressed Berry curvature and quantum metric in solids by optical absorption"

_SciPost Physics, doi:SciPost Phys. Core 5, 040 (2022)_

## Round 2 · Referee Report · Anonymous (Referee 1) · 2022-4-2

Strengths

1 - broad range of application
2 - well detailed equations

Weaknesses

1 - the strength of the formalism should be better explored

Report

In this manuscript, the authors proposed a formalism of quantum metric and Berry curvature for gaped materials at finite-T, including perturbatively the effects of many-body interactions. The formalism relies on the linear response theory of charge polarization, where the Berry curvature and the quantum metric are associated to the dipole energy induced by an oscillating electric field. To demonstrate the proposed formalism, the authors calculated the spectral functions of a disordered Chern insulator. I believe that the manuscript needs to be improved in order to clearly state the novelty, as there are related schemes (approaches) that deals with the same subject.

Requested changes

The manuscript is well written (with very few typos along the manuscript), an with a potential broad range of application in the field. The formalism proposed by the authors is interesting, however most part of the formalism presented is already known in the literature for some time. Indeed, for non-interacting systems, there is even an exercise on the book "Quantum Theory of Electron Liquid" by Giulliani and Vignale to demonstrate the relations between Berry Curvature and linear response functions (although not in the interacting case). In addition, there are criticism on the literature already on the perturbative approach for the non-interacting case, which is mostly used for simple models [see for instance Shin et. al. PNAS March 5, vol. 116, no. 10, 4135–4140 (2019)]. I believe that the authors could use a more complex (and detailed) example to demonstrate the "strength" and novelty of the formalism they are presenting, as there are other schemes where real materials are studied (for instance, exploring the time-evolving Bloch states through the first-principles time-dependent density functional theory).
In summary, I would recommend the authors to clearly demonstrate the novelty of the proposed approach, and show the advantages over existing approaches. This improvement would enhance the manuscript's originality, so that the manuscript would be suitable for publication in SciPost.

  • validity: good
  • significance: good
  • originality: ok
  • clarity: good
  • formatting: excellent
  • grammar: good

Author:  Wei Chen  on 2022-05-24  [id 2518]

(in reply to Report 1 on 2022-04-02)

We thank the referee for the very constructive report, and for recognizing the potentially broad applications of our theory in real experiments. According to the referee's criticisms, we have made a series of modifications to the manuscript. We summarize these changes below, and reply to the referee’s comments point by point.
(1)We have used a single column template in this new submission. This is merely for the sake of organizing the long equations to be in the single column format of SciPost Physics.
(2)We have changed the title of the paper.
(3)We have added the second paragraph to Sec 2.1 to comment on previous linear response theories.
(4)The last sentence of Conclusion is added to mention first principle calculations.
(5)The first paragraph of Sec 2.5 is rewritten to justify the use of disordered Chern insulator as an example.

(Referee's comment)***********The manuscript is well written (with very few typos along the manuscript), and with a potential broad range of application in the field. The formalism proposed by the authors is interesting, however most part of the formalism presented is already known in the literature for some time. Indeed, for non-interacting systems, there is even an exercise on the book "Quantum Theory of Electron Liquid" by Giulliani and Vignale to demonstrate the relations between Berry Curvature and linear response functions (although not in the interacting case).**************

After reflecting on the referee's criticism and searching for the literature, we agree that various linear response theories have been proposed to relate Berry curvature to certain responses caused by some external fields. Therefore we should not merely advocate the linear response part of our formalism. For this reason we have changed the title of the paper to "Measurement of interaction-dressed Berry curvature and quantum metric in solids by optical absorption". This title emphasizes that we are particularly linking Berry curvature and quantum metric to optical absorption experiments, so it is one particular kind of linear response theory. Moreover we mention "interaction-dressed" and “in solids” to highlight the ubiquity of our formalism in incorporating all kinds of many-body effect that can appear in experimental measurements performed in real materials, which are the advantages that distinguish our theory from previous approaches.
Regarding the linear response theory for Berry curvature, we have found three references, two of which were mentioned by the referee. We comment on these references in the second paragraph of Sec. 2.1, and briefly summarized our understanding below.
(1) The Shin et al 2019 paper investigates the charge and spin currents in the presence of an oscillating vector potential. Because the Berry curvature appears as the anomalous velocity part of the current operator, one can naturally obtain the Chern number after integrating the response over momentum. Likewise, the spin current response also gives rises to the spin Hall conductance that characterizes the quantum spin Hall effect.
(2) Gritsev and Polkovnikov elaborate that Berry curvature can be extracted from the response of the generalized force caused by adiabatically quenching a driving parameter, a phenomenon called dynamical quantum Hall effect.
(3) The quantized Hall conductance as the momentum-integration of Berry curvature, i.e., the well-known TKNN formula, which may be derived from expanding the Bloch state to leading order in the external field, can also be expressed in terms of a frequency-derivative of a linear response function at the zero frequency limit, as stated in the Giuliani and Vignale's book mentioned by the referee. This formula, however, does not seem to directly relate Berry curvature to any experimental observable, but more like a trick to eliminate the frequency dependence in the Lindhard function to obtain the TKNN formula.

(Referee's comment)***********In addition, there are criticism on the literature already on the perturbative approach for the non-interacting case, which is mostly used for simple models [see for instance Shin et. al. PNAS March 5, vol. 116, no. 10, 4135–4140 (2019)].*************

We suppose the referee refers to the statement that “the calculation of Berry curvature and the Chern number for real materials requisitely involves a perturbative approach within linear response theory, which in most cases requires the Wannierization technique to cope with the fine-grid integral over the BZ” in the Shin et al paper, now included as Ref [37]. We agree with the referee that to combine our theory with the band structures obtained from first-principle calculations, there will be a lot more details involved, of which we are currently in collaboration with experts to investigate. For instance, one possible route is to obtain an effective tight binding model for the material at hand via down-folding, and then obtain the conduction and valence band eigenstates accordingly such that the non-Abelian Berry connection in Eq (5) can be defined, and then perform linear response theory and Feynman diagrams to include many-body interactions. In this construction, technical problems like Wannierization can be involved, which may need to be addressed case by case, or one may need to invent some better strategies. As we do not have a definite answer to these technical questions at present, they should be clarified in forthcoming studies. We have added the last sentence in the Conclusions to mention this point.

(Referee's comment)************I believe that the authors could use a more complex (and detailed) example to demonstrate the "strength" and novelty of the formalism they are presenting, as there are other schemes where real materials are studied (for instance, exploring the time-evolving Bloch states through the first-principles time-dependent density functional theory).*************

As mentioned above, we are indeed collaborating with other researchers to investigate how our linear response theory can combine with first principle calculations to incorporate realistic material band structures, which is found to be a very involved subject that will be detailed in a forthcoming study. Nevertheless, we emphasize that we choose to study Chern insulator in a continuum with disorder for the following purposes, which have been included into the first paragraph of Sec. 2.5.
(a) The problem can be analytically solved. How the impurities affect the ARPES line width and subsequently the exciton absorption rate is transparent in this model. All the self-energies of the leading order Feynman diagrams have analytical expressions, and we can see which diagrams are detrimental to the spectral function and which are not. In particular, among the two most frequently used approximations for the exciton absorption rate, we find that the polarization operator using the full Green’s function is the most important, whereas the ladder diagrams can be ignored.
(b) The Chern insulator has topological order, and therefore serves as a very good pedagogical example to study how topological order and quantum geometry are influenced by many-body interactions. Our result points to a very profound statement, namely the insulating gap protects the topological order and quantum geometry against many-body interactions, but only to a certain level because in the small gap region the metric-curvature correspondence in Eq (29), a defining quantum geometrical property of noninteracting Chern insulators, is violated in the presence of disorder.
(c) In fact, because many-body interactions can bring in all kinds of effects to influence the shape of the spectral function, one must first understand all these effects in a simple analytical model before considering more sophisticated factors such as material band structures. We find that this simple model does serve this purpose very well, in the sense that it clearly demonstrates the features mentioned in (a) and (b).

(Referee's comment)**********In summary, I would recommend the authors to clearly demonstrate the novelty of the proposed approach, and show the advantages over existing approaches. This improvement would enhance the manuscript's originality, so that the manuscript would be suitable for publication in SciPost.*********

Below we list explicitly the novelty of our linear response theory in contrast to existing approaches, which have been emphasized in the Conclusions and the first paragraph of Sec. 2.1.
(a) Not only the Berry curvature, the quantum metric and quantum geometric tensor also emerge from the same linear response theory we proposed, and be measured by the same experiment with a different choice of the polarization of the light. As a result, not only the topological properties of a gapped material can be addressed, the quantum geometric properties of the material can also be investigated by our formalism.
(b) We introduce the notion of Berry curvature spectral function and quantum metric spectral function. The significance of these quantities is that they play a role similar to how the ARPES spectral function quantifies the effect of many-body interaction on the Block state. The line width of the ARPES spectral function measures the quasiparticle lifetime, and the shift of the quasiparticle peak quantifies how the band structure is renormalized by interactions. Likewise, the line width of the Berry curvature and quantum metric spectral functions quantifies how the quasiparticle lifetime affects topological order and quantum geometry, and the shift of spectral function peak accounts for the effect from renormalized band dispersion. With these spectral functions at hand, one can then investigate how topological order and quantum geometry are influenced by interactions, such as that demonstrated for disordered Chern insulator in Sec. 2.5, so the range of applications is enormous.
(c) From a many-body theory point of view, to elaborate that these spectral functions can be calculated perturbatively by means of Feynman diagrams is another significant feature of our formalism. These Feynman diagrams connect the Berry curvature and quantum metric to textbook many-body physics technique such as polarization operator, Dyson's equation, Matsubara frequency sum, Bethe-Salpeter equation, etc, as elaborated in Sec.~2.3 and therefore tremendously advance the field of perturbative calculation of topological order and quantum geometry. We are not aware of any previous theory that has expressed the Berry curvature and quantum metric into Feynman diagrams.
(d) Finally, another achievement of our work is to demonstrate that Berry curvature and quantum metric can be defined even at finite temperature. Our theory links the exciton absorption experiments performed at finite temperature, which have been performed since the 1950s, can be used to measure Berry curvature and quantum metric, and the requirement of momentum-resolution has already been achieved about a decade ago by incorporating time-resolved ARPES into pump-probe type of experiments. The experimental result for the spectral function in real materials is predicted to be a very broad line due to many-body interactions, which can still be explained perturbatively by our theory in terms of Feynman diagrams. The fact that we give a physical interpretation to these exciton absorption and pump-probe experiments, and moreover connect them to topological and quantum geometrical properties of real materials, opens up a tremendous range of applications to any semiconductors and superconductors.

Attachment:

many_body_metric_version_25.pdf

---

## Round 3 · Referee Report · Anonymous (Referee 1) · 2022-6-7

Report

Dear editor,

In this revised version the authors have considered my comments from the previous report. I believe that in this newer version the authors have clearly stated the novelty of their work (in face of the already known linear response theories links to Berry curvature), and indeed the title (that was now changed) is more appropriate to the subject that they are discussing. Despite the fact that my suggestion to demonstrate a "more complex and detailed" example were not considered, I believe that the argument used by the authors ("a very involved subject that will be detailed in a forthcoming study") is fine and certainly the readers will be willing to see those results.

In summary, after the improvements done by the authors in this newer version, I would suggest the manuscript for publication in SciPost.

---

## Round 3 · Author Response

Dear Dr. Nancy Sandler,
Our manuscript, with a new title "Measurement of interaction-dressed Berry curvature and quantum metric in solids by optical absorption" has been critically reviewed by a referee from SciPost Physics. While the referee praised the potentially broad applications of our theory, he/she also raised concerns about the novelty of our linear response formalism in contrast to existing approaches in the literature. In this revised version, we have incorporated these concerns and rewritten several parts to emphasize the significance of our formalism, especially the connection to experimental measurable. With these changes, we are confident that the revised manuscript properly replied to the referee's concerns, and hereby resubmit it to SciPost Physics for your consideration.
With best regards,
Wei Chen and Gero von Gersdorff

---

## Round 3 · List of Changes

(1)We have used a single column template in this new submission. This is merely for the sake of organizing the long equations to be in the single column format of SciPost Physics.
(2)We have changed the title of the paper.
(3)We have added the second paragraph to Sec 2.1 to comment on previous linear response theories.
(4)The last sentence of Conclusion is added to mention first principle calculations.
(5)The first paragraph of Sec 2.5 is rewritten to justify the use of disordered Chern insulator as an example.

---

## Editorial Decision

published